# Impact of Interaction between Biochar and Soil Microorganisms on Growth of Chinese Cabbage by Increasing Soil Fertility

Jing Luan, Yang Fu, Wenzhu Tang, Fan Yang [ID], Xianzhen Li [ID] and Zhimin Yu *[ID]

School of Biological Engineering, Dalian Polytechnic University, Dalian 116034, China; luanjing426@163.com (J.L.); fuyang3892642022@163.com (Y.F.); tangwz@dlpu.edu.cn (W.T.); yang_fan@dlpu.edu.cn (F.Y.); xianzhen@mail.com (X.L.)
* Correspondence: yuzm@dlpu.edu.cn

**Abstract:** We investigated the improvement of cabbage growth through the interaction of biochar, which can promote microbial activity, with the microbes in the soil. An increase in cabbage growth could be detected in soil with biochar or soil microbes, but the fresh weight of cabbage in soil supplemented with both biochar and microbes was 8.8% and 5% higher, respectively, than that with either microbes or biochar alone. The phosphorus content in Chinese cabbage was also increased by 36.3% when compared with the control without the addition of biochar and microbes. Such an improvement on cabbage growth is closely related to the soil amelioration. The application of biochar in soil significantly stimulated the growth of soil microbes and further altered the microbial community structure in soil. When 2% biochar and microbes were simultaneously applied in soil, the content of the organic matter and available phosphorus content in soil was 36.7% and 45.5% higher, respectively, than that in soil with or without biochar. The maximal increment in the available potassium content was observed in the soil supplemented with both 5% biochar and soil microbes, which was 46.4% higher than that in soil without the addition of biochar and microbes. Both phosphatase and urease activity in soil were also increased by 61.2% and 49.4%, respectively, by applying 2% biochar in soil together with microbes, as the interaction of biochar with the microbes could promote the activity of soil microbes and enzymes which resulted in an improvement in soil fertility. The interaction of biochar with microbes in soil could promote the plant growth of Chinese cabbage by increasing the soil fertility.

**Keywords:** biochar; cabbage growth; improvement; interaction; soil microbes

## 1. Introduction

Biochar is a solid carbonaceous residue produced by biomass pyrolysis in an oxygen-free or low-oxygen environment and includes charcoal, rice husk charcoal, and straw charcoal. It has a large specific surface area and porosity and strong adsorption and antioxidant abilities [1]. Different raw materials and preparation conditions can lead to changes in the physical and chemical properties of biochar, such as texture, pH, ash content, and nutrient content, resulting in different environmental effects [2]. Numerous studies have shown that biochar has a positive impact on soil quality [3]. The application of biochar can increase the soil's organic matter content, thereby improving soil fertility [4]. According to the raw materials and pyrolysis characteristics used in the production of biochar, it contains nitrogen, phosphorus, potassium, and other nutrients, which are further released into the soil [5]. In addition, biochar can increase soil alkalinity and increase soil cation exchange capacity and conductivity, thereby improving nutrient availability. Biochar significantly increased the bulk density, texture, pore structure, pore shape, aggregate stability, surface area, and water availability of the soil [6]. Thus the soil improvement with biochar has been used to increase agricultural production, whereas most researches on biochar were focused on soil amendment for promoting crop growth [7,8].

Soil microbe is an important participant in soil biochemical process that includes oxidation, nitrification, ammonisation, nitrogen fixation, and sulphuration [9]. It promotes the decomposition of the soil's organic matter and nutrient transformation that has a direct impact on soil fertility [10]. Biochar is a new soil improvement material that can change the number and structure of microbial populations in soil and enhance its function. The specific effects of biochar on microbes can be divided into several categories [11]: (1) biochar with high surface area and porous structure serves as a shelter for soil microbes; (2) biochar provides nutrients such as carbon and nitrogen sources for soil microbes; and (3) biochar improves soil properties and promotes microbial growth. Biochar was able to regulate the microbial community structure and activity through a differential impact on soil microbes in the process of biochar amending soil. Such alteration is beneficial to the construction of the soil environment being conducive to plant growth and increasing soil fertility for crop production [12]. However, the impact of the interaction between biochar and soil microbes on plant growth is rarely reported. Soybean growth could be improved in the biochar-amended soil, and the microbial diversity in the rhizospheric soil was significantly increased, most of which have the characteristics of promoting plant growth and resisting against phytopathogen [13]. Yang et al. [14] reported that corncob biochar could improve potassium-/phosphate-solubilising activities by promoting the bacterial growth of *Ochrobactrum* sp. and *Bacillus mucilaginosus*.

According to the available information, there is no investigation on the promotion of plant growth by the interaction of biochar with microbes to enhance soil fertility. Therefore, the aim of this study was to investigate the effect of the interaction of biochar with phosphorus-/potassium-solubilising bacteria on soil fertility. The other aim was to examine the influence of the interaction between biochar and soil microbes on cabbage growth and quality. The third and the final aim was to explain the impact of interaction between biochar and soil microbes on cabbage growth by increasing soil fertility. This will be beneficial in helping to understand the mechanism of biochar increasing crop yield.

## 2. Materials and Methods

### 2.1. Soil, Biochar, Chinese Cabbage and Chemicals

A top soil layer taken from a 0–20 cm depth was collected for pot experiments from farmland in southern Liaoning, China (39°32′ N and 122°0′ E). This field was a kind of brown earth, in which a maize crop has been sown and harvested for 3 years. The soil sample was pH 7.0, its electrical conductivity was 550.5 $\mu$S cm$^{-1}$, and it contained total nitrogen (3.6 g kg$^{-1}$), hydrolyable nitrogen (80.1 mg kg$^{-1}$), available phosphorus (9.8 mg kg$^{-1}$), available potassium (59.1 mg kg$^{-1}$), and organic matter (9.4 g kg$^{-1}$). The soil was autoclaved at 0.1 Mpa pressure and 121 °C temperature for 45 min using the high-pressure steam steriliser (Shanghai Shen'an Medical Device Factory, Shanghai, China).

The biochar was prepared using the oxygen-limited pyrolysing method, according to the described method by Chan et al. [15]. The corncob was oxygen-limited pyrolysised at 450 °C for 4 h and lightly crushed and sieved to obtain a uniform particle size of 0.25–2.0 mm. The biochar was pH 9.8, its electrical conductivity was 550.5 $\mu$S cm$^{-1}$, its specific surface was (60.7 m$^2$ g$^{-1}$), and its aperture was (0.06 cm$^3$ g$^{-1}$). It contained organic matter (88.8 g kg$^{-1}$), total nitrogen (8.5 g kg$^{-1}$), total phosphorus (2.4 g kg$^{-1}$), and total potassium (18.0 g kg$^{-1}$).

The Chinese cabbage was purchased from a local supplier. Potash feldspar powder and phosphorus ore powder were purchased from Dabieshan Mining Co., Xinyang, China, which was sieved at a 0.149 mm size and washed with distilled water. All the reagents were purchased from Shenggong Co., Shanghai, China.

### 2.2. Strains and Culture Condition

*Ochrobactrum* sp. ACCC10085 was used as a phosphorus-solubilising bacterium and purchased from the Agricultural Culture Collection Centre of China. *Bacillus mucilaginosus*

AS1.153 was used as a potassium-releasing bacterium and purchased from the China General Microbiological Collection Centre.

*Ochrobactrum* sp. ACCC10085 and *B. mucilaginosus* AS1.153 were cultured in the nutrient broth at 30 °C and 150 rpm until the $OD_{600}$ reached approximately 1.0. The culture was collected for the experiment. The nutrient broth constituted (per litter) beef extract (3 g), peptone (5 g), and NaCl (5 g) at pH 7.0.

### 2.3. Experimental Design

The sterilised soil sample (2 kg) was filled in a pot (20 cm × 20 cm in height and diameter), in which biochar, both *Ochrobactrum* sp. ACCC10085 and *B. mucilaginosus* AS1.153, and potash feldspar plus phosphorus ore were mixed thoroughly. The pot experiments comprised 8 treatments including biochar application at 0, 0.5%, 2%, and 5%, with or without 5% cultures of *Ochrobactrum* sp. ACCC10085 and *B. mucilaginosus* AS1.153 and 1% powder of potash feldspar and phosphorus ore together. Each treatment was designed with four replicates. The soil in the pot was moistened to 50% water-holding capacity and kept for 1 day in a greenhouse. Cabbage seeds were sown in pot and thinned to four plants each pot after germination.

### 2.4. Soil Measurement

At end of the growth period, soil samples were collected from the cabbage rhizosphere. Microbial biomass in soil was measured by the colorimetric 3-(4,5-Dimethyl-2-thiazolyl)-2,5-diphenyltetrazolium bromide (MTT) method as described before [16]. The soil's organic matter was determined by wet digestion with $K_2Cr_2O_7$ oxidation [17]. The hydrolyable nitrogen content in soil was determined with the Conway method [17]. The available potassium content in the soil was determined using an atomic absorption spectrometer (Z-8100, Hitachi, Tokyo, Japan) [16] and the available phosphorus content in the soil was determined using the molybdenum blue method [18]. Urease activity in the soil was determined using the sodium phenol-sodium hypochlorite colorimetric method [18]. Phosphatase activity in soil was determined using the disodium phenylphosphate colorimetric method [18].

### 2.5. Plant Measurement

After sowing for 35 days at room temperature, the Chinese cabbage was harvested. The residual soil on the root system was washed out and the whole plant was weighed as the wet weight of cabbage. After it was dried to a constant weight at 105 °C, the plant was ground up and sieved using a particle size of 0.18 mm. The phosphorus and potassium contents in the cabbage tissue were assayed as described in soil measurement.

### 2.6. Determination of Soil Bacterial Diversity

The genomic DNA from the soil samples was extracted using a DNA extraction kit (Omega Bio-tek, Norcross, GA, USA). Using the genomic DNA as a template, selecting bacterial diversity identification region 16SV3-V4 (primers: 343F sequence 5′-TACGGRAG-GCAGCAG-3′and 798R sequence 5′-AGGGTATCTAATC-CT-3′) was amplified by two rounds of PCR to obtain raw data. After using Trimmatic software (v 0.38) to remove impurities from both ends of the original sequence, FLASH software (v 1.2.11) was used for splicing. Then, Vsearch software (v 2.8.1) was used to classify the sequence into multiple OTU units based on its similarity. The QIIME 2 software package was used to select representative sequences for each OTU, and all representative sequences were compared and annotated with the database.

### 2.7. Statistical Analysis

All tests were carried out triplicate. The test values were expressed as the mean ± standard deviation. Analysis of variance and significant differences among means were tested by an independent-sample *T* test ($p < 0.05$) using SPSS software (version 22.0, SPSS Inc., Chicago, IL, USA). The Pearson test (two-tailed) was used to analyse whether and

how cabbage growth, the soil's organic matter, available phosphorus content, available potassium content, and soil enzyme activity were affected by the interaction of biochar with microbes using SPSS software (version 22.0, SPSS Inc.).

## 3. Results

### 3.1. Promotion to Cabbage Growth by the Interaction of Biochar with Microbes

After being grown in soil that has had different treatments, the Chinese cabbage was harvested and its fresh weight was determined (Figure 1). When compared with the control grown in the sterilised soil, there were some differences in the fresh weight in Chinese cabbages grown in different soil treatments. The wet weight was increased by 4.2%, 11.6% and 13.6%, respectively, with the increasing addition of biochar when compared with that in soil without the application of biochar. Such an increase was also observed in the sterilised soil with the addition of microbes. However, when microbes were applied together with biochar in soil, its increment of the wet weight was higher than that when only biochar or microbes were applied in soil. The cabbage growth was increased maximally by 5% in soil with both biochar and microbes than that with biochar solely and by 8.8% than that with microbes alone. It appeared that the interaction of biochar with microbes promoted cabbage growth stronger than biochar or microbes alone.

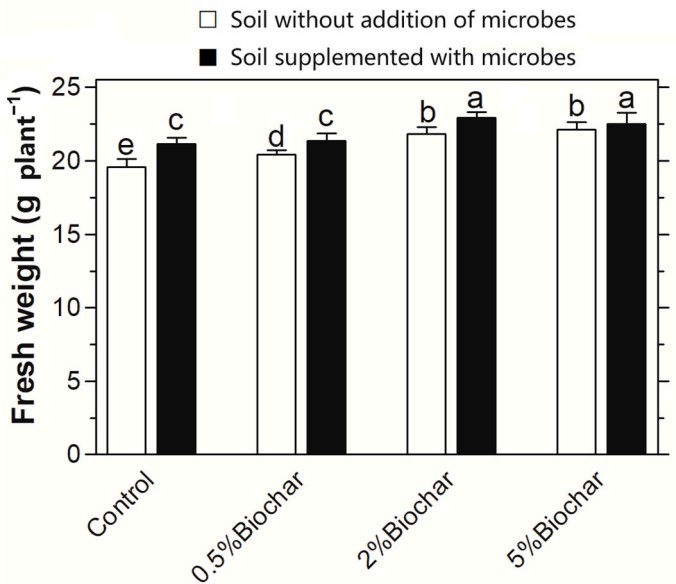

**Figure 1.** Influence of the interaction between biochar and soil microbes on the plant growth of Chinese cabbage. Different letters on the top of bars indicated significant differences among the different treatments ($p < 0.05$).

### 3.2. Biochar Stimulates Microbial Activity in Soil

In order to investigate the influence of biochar on microbial activity in soil, both *Ochrobactrum* sp. ACCC10085 and *B. mucilaginosus* AS1.153 were applied in the soil with addition of biochar. After the Chinese cabbage was sowed for 35 days the microbial biomass in the rhizosphere soil was measured (Figure 2). The application of biochar in soil significantly stimulated microbial growth, in which the addition of biochar at 2% showed a maximal promotion to microbial growth. The microbial biomass was maximally increased by 120% when compared with the control without the addition of biochar. However, the addition of biochar at 5% showed a significant inhibitory effect on microbial growth.

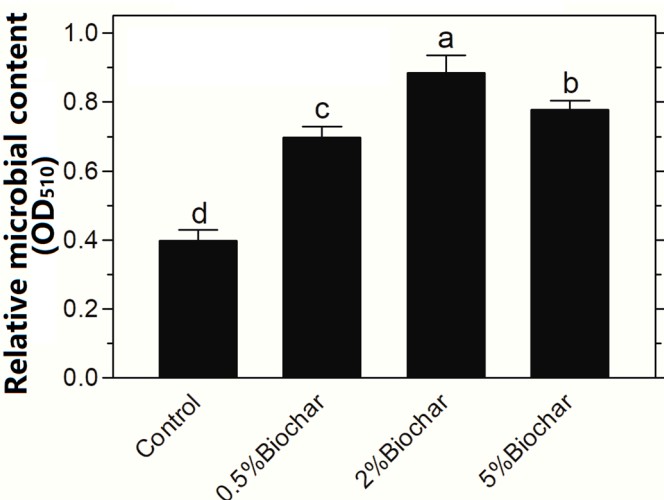

**Figure 2.** Effect of biochar addition in soil on microbial growth. Different letters on the top of bars indicated significant differences among the different treatments ($p < 0.05$).

Figure 3 showed that the results of the relative abundances of bacterial communities at the genus levels. In the group of the addition *Ochrobactrum* sp. ACCC10085 and *B. mucilaginosus* AS1.153, *Ochrobactrum*, *Bacillus*, *Pseudomonas*, *Pantoea* and *Chryseobacterium* dominated the bacteria genus with 31.1%, 20.1%, 10.4%, 7.3%, and 5.5% of the entire bacterial communities, respectively. After the application of 0.5–2% biochar, the proportion of *Ochrobactrum* increased in soil samples after 35 days of cabbage growth ranging from 29.4% to 47.1%, compared with the samples that only had the addition of microbes. When the addition mass of biochar was increased to 5%, the proportion of *Bacillus* in soil supplemented with both biochar and microbes was increased, which was 60.6% higher than that of the control samples.

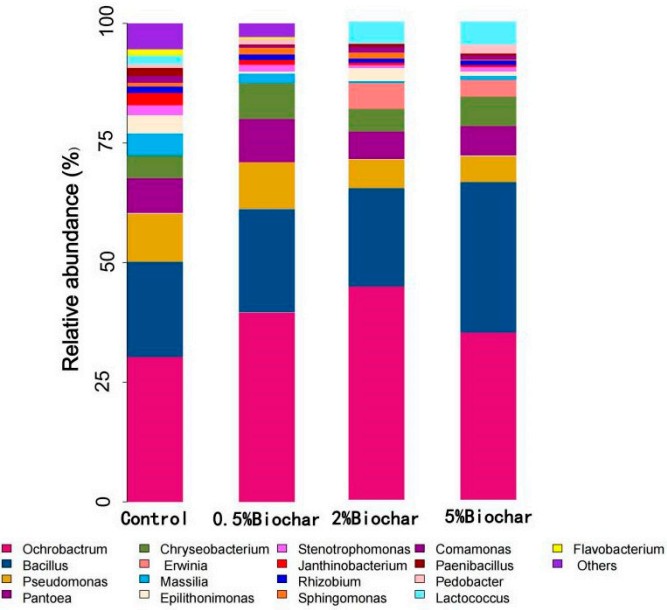

**Figure 3.** Effect of biochar on relative abundances of microbial community at the genus levels.

### 3.3. Improvement of Soil Fertility by the Interaction of Biochar with Microbes

After the Chinese cabbage was sowed in soil for 35 days, the rhizosphere soil was assayed for the soil's organic matter (Figure 4A). The content of the organic matter in the soil was increased with the increasing addition of biochar. When 5% biochar was added to the soil, the soil's organic matter content reached a maximum of 12.1 mg g$^{-1}$. On this

basis, when microbes were applied together with biochar in the soil, the content of the soil's organic matter was further increased compared to that when only biochar was applied to the soil. When the microbes of *Ochrobactrum* sp. and *B. mucilaginosus* were applied to the soil supplemented with 2% biochar, the the soil's organic matter was highest and was 36.7% higher than the control in the sterilised soil, as well as 7.0% higher than that in the soil with 2% biochar. Thus, the accumulation of organic matter in soil with both biochar and microbes was higher than that in soil with biochar alone. However, when 5% biochar and soil microbes were added to the soil, the content of the organic matter was significantly reduced, which was 16.3% lower than that in soil with 2% biochar and soil microbes.

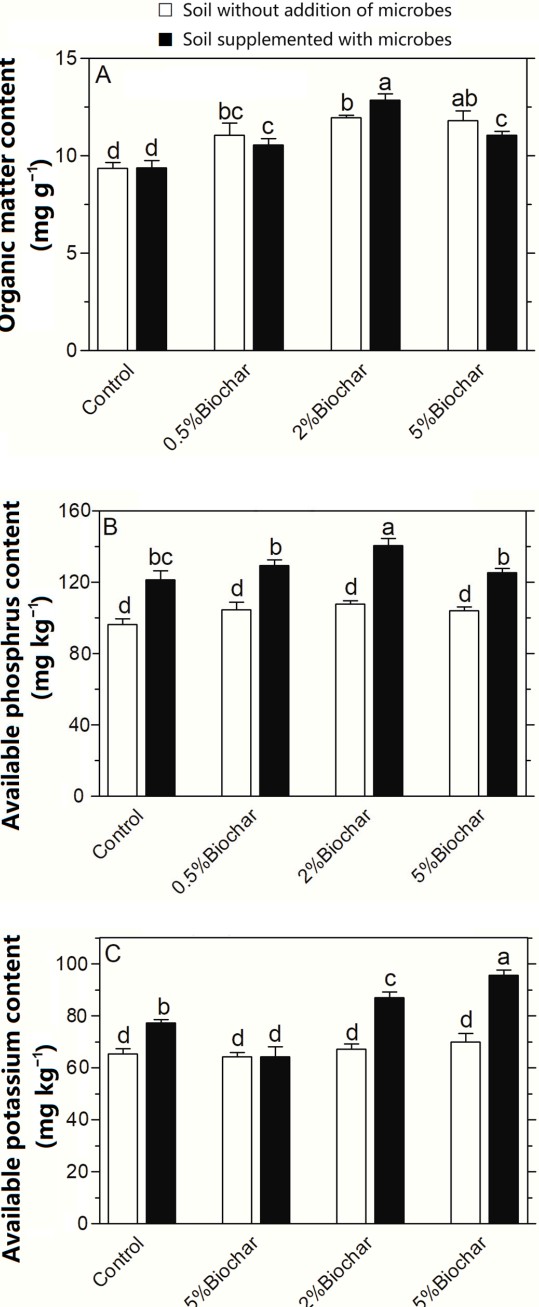

**Figure 4.** Improvement of soil fertility including the soil's organic matter (**A**), available phosphorus (**B**) and available potassium (**C**) by the interaction of biochar with soil microbes in soil. Different letters on the top of bars indicated the significant differences among the different treatments ($p < 0.05$).

The available phosphorus/potassium contents were compared in different soil treatments. As shown in Figure 4B, there was no significant difference in the available phosphorus content regardless of whether biochar was applied to the soil or not, but the available phosphorus content in soil was increased significantly when the microbes of *Ochrobactrum* sp. and *B. mucilaginosus* were applied to the soil. The maximal increment in the available phosphorus content was observed in the soil supplemented with 2% biochar and soil microbes, which was 45.5% higher than that in the sterilised soil without the addition of biochar and microbes. Its content in soil with both 2% biochar and microbes was 30% higher than that in soil with 2% biochar, and was 15.4% higher than that in soil with microbes. The interaction of biochar with microbes promoted the phosphorus-releasing activity of *Ochrobactrum* sp. ACCC10085 causing the increase in phosphorus content. However, when 5% biochar and soil microbes were added to the soil, the content of the available phosphorus in the soil was significantly reduced, which was 10.0% lower than that in soil with 2% biochar and soil microbes.

To examine the influence of the interaction between biochar and microbes on the available potassium content in soil, both biochar and microbes were applied to the soil and the available potassium content was assayed (Figure 4C). Significant differences in the available potassium content were not detected in soil with varying amount of biochar, whereas the available potassium content was significantly increased when the microbes of *Ochrobactrum* sp. and *B. mucilaginosus* were applied in the soil. The maximal increment in the available potassium content was observed in the soil supplemented with both 5% biochar and soil microbes, which was 46.4% higher than that in the soil without the addition of biochar and microbes. The content of soil-available potassium with both 5% biochar and microbes showed increases of 37.6% and 23.6%, respectively, compared to that in soil with 5% biochar alone and microbes alone.

*3.4. Improvement of Soil Enzymes Activity by the Interaction of Biochar with Microbes*

The activity of phosphatase and urease was compared in different soil treatments. As shown in Figure 5A, biochar could significantly increase the acid phosphatase activity in the soil. When microbes were applied together with biochar in the soil, the acid phosphatase activity was increased compared to that when only biochar or microbes were applied to the soil. When the microbes were applied in soil supplemented with 2% biochar, the acid phosphatase activity was highest and 61.2% higher than that of control in the sterilised soil, as well as 39.3% higher than that in soil with 2% biochar.

As shown in Figure 5B, significant differences in urease activity were not detected in soil with increasing amounts of biochar, whereas urease activity was significantly increased when the microbes were applied to the soil. The maximal increment in urease activity was showed in the soil added with both 2% biochar and soil microbes, which was 49.4% higher than that in the soil without the addition of biochar and microbes. There was a significant reduction of urease activity, when the biochar mass reached 5%.

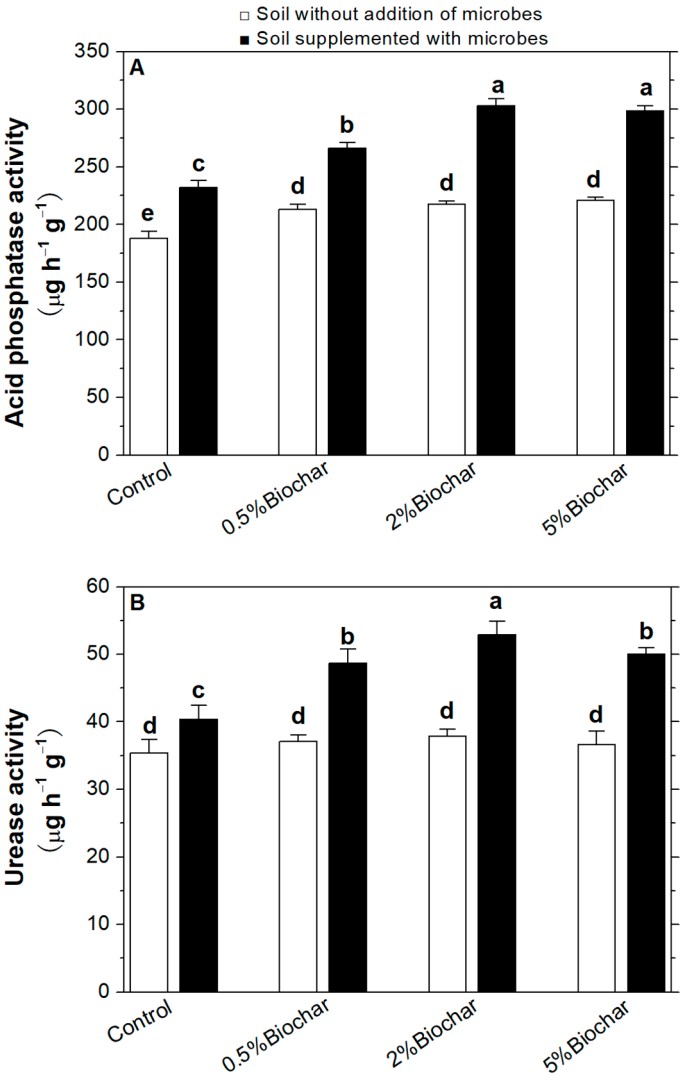

**Figure 5.** Improvement of soil phosphatase activity (**A**) and urease activity (**B**) by the interaction of biochar with soil microbes in soil. Different letters on the top of bars indicated the significant differences among the different treatments ($p < 0.05$).

*3.5. Cabbage Improvement by the Interaction of Biochar with Microbes*

The phosphorus and potassium contents in the Chinese cabbage were assayed to evaluate the influence of the interaction between biochar and microbes on cabbage quality (Figure 6). Phosphorus content in cabbage was significantly increased in soil supplemented with biochar and/or microbes when compared with the control of soil without biochar and microbes (Figure 6A). It was 36.3% higher in soil with both biochar and microbes than that without the addition of biochar and microbes, suggesting that the interaction of biochar with microbes was capable of improving cabbage quality regarding its phosphorus content. However, there was no significant difference in the cabbage's potassium content regardless of whether biochar and/or microbes were applied to the soil or not (Figure 6B).

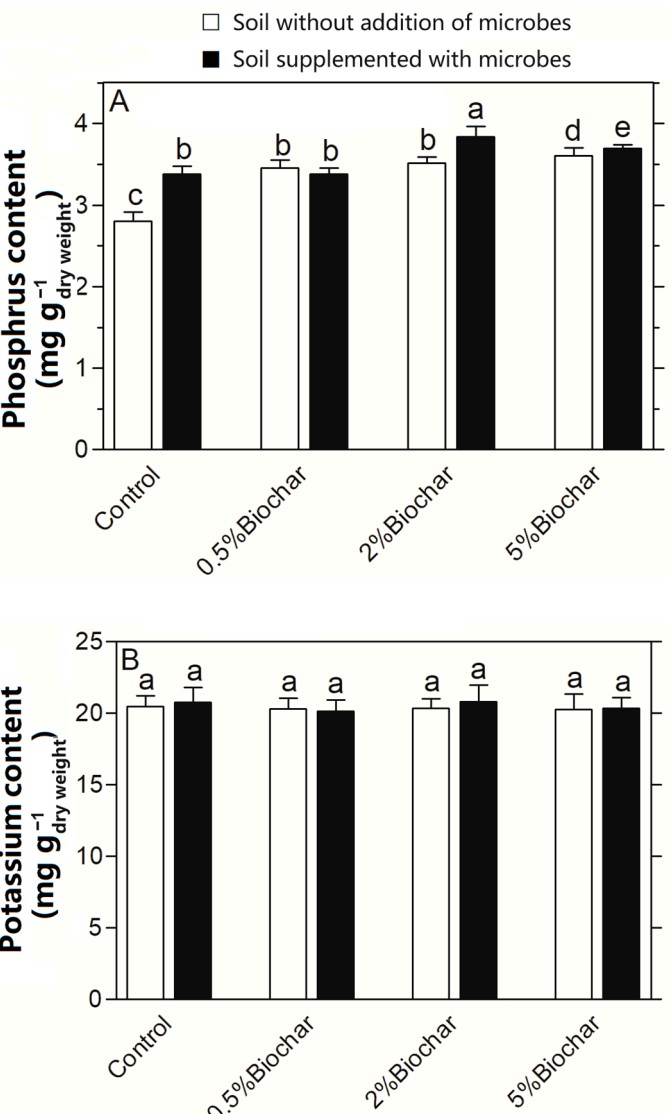

**Figure 6.** Influence of the interaction between biochar and soil microbes on the contents of phosphorus (**A**) and potassium (**B**) in Chinese cabbage. Different letters on top of bars indicate significant differences among the different treatments ($p < 0.05$).

### 3.6. Relationships between the Interaction of Biochar with Microbes and the Properties of Soil and Cabbage Growth

Pearson correlation analysis revealed the correlation between the growth of soil microbes (microbial content, *Ochrobactrum* sp. relative abundance and *Bacillus mucilaginosus* relative abundance) and the properties of soil and cabbage growth (Table 1). The growth of soil microbes was positively correlated with all 8 indices of soil and cabbage growth (i.e., OM, AP, AK, PS, US, FW, PC, and KC). There was a significant positive relationship between the microbial content, the soil properties (OM, AP, PS, and US) and cabbage growth (FW and PC). Similarly, the relative abundance of *Ochrobactrum* sp. showed a significant positive correlation with other indicators, except for a non-significant correlation with AK. The relative abundance of *Bacillus mucilaginosus* was only significantly correlated with the related potassium indicators (AK and KC).

**Table 1.** Pearson correlations between the interaction of biochar with soil microbes and the properties of soil and cabbage growth after cabbage harvest (*n* = 12) [a].

| Indexes [b] | Microbial Content | The Relative Abundance of *Ochrobactrum* sp. | The Relative Abundance of *Bacillus mucilaginosus* |
|---|---|---|---|
| OM | 0.952 * | 0.987 * | 0.657 |
| AP | 0.987 * | 0.991 ** | 0.501 |
| AK | 0.657 | 0.664 | 0.959 * |
| PS | 0.998 ** | 0.977 * | 0.636 |
| US | 0.975 * | 0.976 * | 0.494 |
| FW | 0.958 * | 0.973 * | 0.820 |
| PC | 0.970 * | 0.990 ** | 0.755 |
| KC | 0.945 | 0.984 * | 0.965 * |

a: ** $p < 0.01$; * $p < 0.05$. b: OM: Organic matter in soil; AP: Available phosphorus in soil; AK: Available potassium in soil; PS: Phosphatase activity of soil; US: Urease activity of soil; FW: Fresh weight of cabbage; PC: Phosphorus content of cabbage; KC: Potassium content of cabbage.

## 4. Discussion

That Biochar can promote crop growth and performance when added to soil has been previously reported for crops such as soybean [8], maize [1], and cabbage [19]. It has also been known that soil microbes can promote plant growth by soil amendment [20]. However, there was less information about the impact of the interaction between biochar and microbes on plant production [21]. Thus, biochar was applied together with microbes in the soil planted with Chinese cabbage. In the present study, we found that the interaction of biochar with microbes promoted cabbage growth stronger than biochar or microbes alone, presumably that the biochar interaction with microbes in soil could enhance crop growth and performance. These outcomes confirmed previous findings that indicated the positive effects of biochar–microbe interactions after the combination of biochar and microbe application in soil on the growth, yield, and nutrient availability of different plant species such as soybean [8], sunflower [22], and rice [23]. The improved examined growth parameters as a result of biochar–microbe interactions might be ascribed to the role of biochar in microbial growth and activity in soil stimulation in general [24]. It is well known that soil microbes are an important soil component of ecosystems, playing an important role in soil organism transformations, the maintenance of soil fertility, and plant material conversions [4].

Soil microbes are active contents that can improve soil fertility by impacting on the microbial community structure in rhizosphere soil and its nutrient cycling [25]. Previous studies suggested that soil microbial biomass could be increased in soil supplemented with biochar and that the microbial community structure and abundance were changed over time [26]. Both soil microbes, *Ochrobactrum* sp. ACCC10085 and *B. mucilaginosus* AS1.153, were applied in the soil with the addition of biochar in the present study. The microbial content in the rhizosphere soil was higher when biochar was added. Biochar has a porous structure and the ability to change soil porosity [27], which might have created a better environment for soil microbes to grow and reproduce. Considering that these two microbes have an ability to solubilise phosphorus and potassium, it was presumed that the interaction of biochar with the microbes could improve the soil fertility by enhancing microbial activity [13].

The impact of biochar on soil microbes is diverse. Many studies provided different hypotheses to explain the effect of biochar application on soil microbe growth and activity. Leal et al. [28] and Zhang et al. [29] reported that biochar may increase the soil bacteria amount by improving the soil's pH. Gul et al. [9] concluded that the high cation exchange capacity and adsorption capacity of biochar helped to maintain nutrients in the soil and provide substrates for the growth and metabolism of soil microbes. Asfaw et al. [7] found a high root colonisation level of arbuscular mycorrhizal fungi as a result of biochar application. In addition, the biochar indirectly affected the growth and metabolic activity of soil microbes by altering the soil properties, regulating nutrient availability, and regulat-

ing enzyme activity [30]. Overall, our results demonstrate the potential role of biochar application in improving soil microbe growth and reproduction in soil.

However, adding excessive biochar to the soil can also have an inhibitory effect on soil microbes (Figure 2). Excessive biochar adsorbs nutrients and minerals from the soil, which limits the nutrition of soil microbes and affects their growth [31]. On the other hand, the potential toxicity caused by the emission of volatile pyrolysis products and water-soluble extracts play a leading role in affecting the biomass and composition of soil microbial communities. Sun et al. [32] reported that biochar released high concentrations of phenolic compounds that decreased the number of *Bacillus mucilaginosus* in soil. Bueno et al. [33] reported that the biochar components extracted by water in pyrolysis sugarcane and corn cob inhibited the growth of *Leocobacter* sp. and *Bacillus aryabhatai*. Therefore, adding biochar to soil requires consideration of its amount. An appropriate amount of biochar can promote microbial growth, while an excessive amount of biochar can have an inhibitory effect on soil microbes. The optimal amount of biochar added in this study is 2% (Figure 2). Similar results were found by others authors. Karimi et al. [34] reported that the application of 1% and 2% (*w/w*) corn residue biochar in calcareous soil increased soil microbial biomass by 20% to 124% compared to that of the control.

Adding biochar to soil can alter the living environment of microbes, leading to an increase in the abundance of certain dominant bacteria or inhibition of the growth of certain pathogens. Fox et al. [35] reported that biochar significantly altered the abundance of rhizosphere microbes, with an increase of 100 times in the abundance of phosphorus solubilisation-related aeromonads, sulphate desulfurization-related arthrobacter, and copper-loving bacteria. Wang et al. [36] showed that the relative abundance of fungi and bacteria, as well as the content of PLFA, decreased with the increase of corn biochar application. This can be explained by the alkalinity of biochar, which often inhibits soil microbial activity in the short term, thereby reducing community composition. In this study, *Ochrobactrum* sp. ACCC10085 and *B. mucilaginosus* AS1.153 strains were added to the soil, which became dominant strains in the soil. When adding 0.5–2% biochar, the proportion of the *Ochrobactrum* sp. ACCC10085 strain in the soil was relatively high (Figure 3), and the proportion of the *B. mucilaginosus* AS1.153 strain in the soil was relatively high when adding 5% biochar (Figure 3).

Biochar was reported to increase crop productivity accompanied by an improvement in soil fertility [37]. Numerous studies have shown that biochar has a positive impact on soil quality. Applying biochar can increase the soil's organic matter content, thereby improving soil fertility [38]. According to the raw materials and pyrolysis characteristics used in the production of biochar, it can contain some nutrients, but due to its highly porous structure, it can improve the retention of nutrients in the soil. In addition, biochar can reduce soil acidity and increase soil conductivity and cation exchange capacity, thereby improving nutrient availability [6].

The soil's organic matter is an important indicator for soil fertility that mainly comes from organic fertiliser and the residual root and root exudates of the crops in soil [20]. The changes in the soil were examined in different soil treatments to make clear whether the interaction of biochar with microbes can enhance soil fertility. In the present study, we observed that the accumulation of organic matter in soil with both biochar and microbes was higher than that in soil with biochar alone and that presumably the interaction of biochar with microbes could enhance the transformation of the soil's organic matter. This could be because some components in biochar, including minerals, volatile organic compounds, and free radicals, can potentially influence microbial activity, reshape the soil's microbial community, and change the soil's enzyme activity that catalyses various key biogeochemical processes, including the soil's organic matter content and elemental (e.g., N, P, and K) cycles [20]. In addition, due to the strong adsorption performance of biochar, it also increases the complexity of soil enzymes. The adsorption of biochar on reaction substrates facilitates enzyme reactions and promotes the soil's enzyme activity, as well as increasing the soil's organic matter content.

However, when the amount of added biochar is too high (5%), the organic matter content in the soil significantly decreases (Figure 4A). This may be due to the excessive toxic substances (heavy metals, polycyclic aromatic hydrocarbons, etc.) that spread into the soil from the biochar that inhibits the growth of soil microbes (Figure 2), which reduce the soil's enzyme quantity and the release of the soil's organic matter. These toxic substances also have an inhibitory effect on the soil's enzyme activity. In addition, Bailey et al. [39] reported that the adsorption of soil enzymes on biochar can reduce the availability of binding sites, which inhibit enzymatic reactions and the soil's organic matter release.

Soil microbes are an important factor affecting soil fertility, and biochar could affect the growth activity of some phosphorus-/potassium-releasing bacteria [14]. In the present study, we found that significant increases of available phosphorus/potassium content in soil with both biochar and microbes when compared with the levels found in soil with either biochar or microbes alone. Presumably, the interaction of biochar with microbes could promote microbial phosphorus-/potassium-solubilising activity, which caused an increase in the phosphorus/potassium content in the soil. The proportion of the *Ochrobactrum* sp. ACCC10085 strain in the soil with the addition of 0.5–2% biochar was relatively high, which could promote soil phosphorus dissolution and increase the available phosphorus content in the soil. However, the proportion of the *Ochrobactrum* sp. ACCC10085 strain in soil with the addition of 5% biochar was decreased, which led to a decrease in phosphorus solubility as well as the available phosphorus content in the soil. This strain content was closely related to the available phosphorus content in the soil (Table 1). The proportion of the *B. mucilaginosus* AS1.153 strain in soil with the addition of 5% biochar was relatively high, which could promote the dissolution of soil potassium and increase the content of available potassium in the soil. The bacterial content was closely related to the available potassium content in the soil (Table 1).

Biochar affects the activity of extracellular enzymes in soil, which are responsible for the degradation of organic carbon and the activity of other important phosphorus-solubilising enzymes or potassium lyase. Soil enzyme activity is influenced by factors such as the pH, cation exchange capacity, water capacity, and pore structure. Ameloot et al. [40] found that high and low temperature biochar can cause different changes in enzyme activity. Research has shown that the impact of biochar on the soil's enzyme activity varies depending on the type of biochar and soil enzyme type. Biochar contains phosphorus, potassium, magnesium, and other nutrients, which can promote soil microbial activity and thereby improve soil fertility. It was concluded that the interaction of biochar with soil microbes could enhance soil fertility, including organic matter and the phosphorus and potassium content that was able to promote the plant growth of Chinese cabbage.

Phosphorus accumulation in Chinese cabbage was significantly higher in the combined treatments of biochar and soil microbes than in the control (Figure 6A). Enhanced phosphorus accumulation by plants as a result of biochar application could be attributed to the high phosphorus content of the corncob biochar and thus higher phosphorus content and availability in biochar-amended soils [41]. Improved soil pH and cation exchange capacity, leading to better nutrient absorption and reduced phosphorus fixation, could also be the case [20]. Biochar is known to facilitate the growth and activities of phosphorus-solubilising microbes, which mobilise phosphorus for uptake by plant roots [42]. Increase in soil pH and soil microbes biomass from biochar application might contribute to the increase in plant phosphorus accumulation. Fox et al. [24] have also observed an increase in the phosphorus concentration in plants after the interaction of biochar with microbes.

## 5. Conclusions

When microbes were applied to soil together with biochar, the content of the organic matter in the soil was higher than that in soil with or without the addition of biochar, and an increase in the phosphorus and potassium contents was obtained in soil with both biochar and microbes when compared with that of soil without the addition of biochar and/or microbes. Such increments were the result of the promotion of microbial activity

by the interaction of biochar with microbes in soil, which caused an improvement in the soil's fertility. Thus, the interaction of biochar with microbes could promote the growth of Chinese cabbage by stimulating microbial activity. The phosphorus content in the cabbage was also increased by the interaction between the biochar and microbes in the soil to improve cabbage quality.

**Author Contributions:** Conceptualization, Z.Y.; methodology Y.F.; software and formal analysis, F.Y.; writing-original draft preparation, J.L.; writing-review and editing, W.T. and X.L. All authors have read and agreed to the published version of the manuscript.

**Funding:** This study was founded by the Basic Scientific Research Projects of Liaoning Education Department (LJKQZ2021114); Dalian Key Technology Research and Development Plan (2022YF16SN064).

**Institutional Review Board Statement:** Not applicable.

**Informed Consent Statement:** Not applicable.

**Data Availability Statement:** All data produced or examined during this research can be found in the article. Further inquiries can be directed to the corresponding author.

**Conflicts of Interest:** The authors declare no conflict of interest.

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
