# Peer review of "Impact of Interaction between Biochar and Soil Microorganisms on Growth of Chinese Cabbage by Increasing Soil Fertility"

_applsci, doi:10.3390/app132312545_

Round 1

Reviewer 1 Report

Comments and Suggestions for Authors

The authors have investigated the impact of biochar and soil microorganism on the growth of Chinese Cabbage. The results indicate improved plant growth when both biochar and microorganisms used. In spite of the fact that the study is interesting, a deeper understanding of the mechanism of the actual improvement in performance is lacking. Rather than simply mentioning observations, a more detailed discussion is required to support the claim.

1) "Soil sample was sterilized at 121 degC for 45 min before pot experiment." The sentence must be elaborated. The readers must understand how the complete sterilization process was carried out. It helps to replicate the experiments.

2) Fig.2 What is the reason for a decrease in the microbial content for 5% Biochar? The authors must discuss it in the revised manuscript.

3) Same applies in Fig. 3 (a), (b). The author must add a discussion on the reduced organic matter content and Phosphorous content.

4) Why the Potassium content using 5%biochar content is higher ?

5) check the units in the graphs. "mg" or "g"

Reviewer 2 Report

Comments and Suggestions for Authors

Dear Editor and authors, the manuscript has the potential to be published, but as it is structured and written it should not be. Authors need to make many changes before the article can be accepted. Below are some suggestions:

- The Introduction is poorly written and needs to be redone, highlighting the aims of the study, the treatments tested and the main results.

- The introduction is poor, the authors should provide more background on the interaction between microorganisms and biochar, as despite being a new area of study there are already several works on it, the authors need to improve their bibliographic review on the topic.

- Furthermore, the results section at the end of the introduction should be removed, and the manuscript hypothesis and aims should be provided at the end of the introduction.

-The material and methods section must be rewritten in order to ensure the reproducibility of the experiment, the authors omit a lot of important information, such as biochar property and pyrolysis conditions, as well as soil chemical properties, the authors need to provide more information. Authors must be guided by scientific writing styles of articles already published in this journal.

- The methodologies of the elements evaluated in the soil, is it to estimate the available or the total? – the authors need to provide more details on the entire study methodology.

- In the results section, in the figures, authors must insert a caption within the figure to explain what the two colors mean, and not provide this information only in the figure description.

- The type of statistical approach is not the best used, the authors created a factorial scheme, but when comparing treatments they ignore this, so all statistical analyzes must be redone.

- Authors must perform a correlation analysis or principal components analysis relating the measured variables to establish a cause and effect relationship that will help in discussing their data.

- The discussion section must not be divided into topics, it must be continuous and the previous paragraph must be aligned in the discussion with the following one.

- The discussion is reasonable, but the authors need to highlight which mechanisms were observed in the study and go deeper into them. The authors mention that biochar has conditions that favor the development of microorganisms, and discuss this with their data, however, this is done speculatively, with correlation analysis and providing the properties of biochar it will be easier and more assertive to delimit the mechanisms that acted in their study.

Reviewer 3 Report

Comments and Suggestions for Authors

Comments: This manuscript “Impact of interaction between biochar and soil microorganisms on growth of Chinese cabbage by increasing soil fertility” presented the effects of simultaneous application of both biochar and microbes on growth of Chinese cabbage. The manuscript was well written with previously well-known logical flow with little to no novelty in the manuscript. The authors did not characterize (XRD, XPS, FTIR etc) the biochar prepared and the basic physical and chemical properties of biochar and soil used were not presented. Also, the workload of the manuscript is very small and the authors need to extensively develop and improve every aspect of the manuscript. The research idea was great; however, the planning, execution and scope were too insignificant. The methodology was not up to standard and most discussion of results are too shallow and unconvincing and need to be back up with authentic and reasonable scientific backing/references. Lastly, the similarity index of the manuscript is relatively too high.

Abstract:

1.     In the abstract provide the percentage increment among the treatments for all major parameters measured in this research and add more results.

Introduction:

1.     The authors should add more contents about Biochar, e.g Biochar is a highly porous, carbon-rich substance created through the pyrolysis of organic matter…… Also, talk about the beneficial effects of biochar to soil and plants in the introduction. Also, introduce the various mechanism of action of biochar. Consider the following references and use them to improve your work: 10.1016/j.jplph.2023.154023; 10.1080/03650340.2022.214610010.1007/s11104-023-06256-4.

Methodology:

1.     Why did you choose this percentage?   0, 0.5%, 2% and 5% with or without 5% cultures of Ochrobactrum sp. ACCC10085 and B. mucilaginosus AS1.153

2.     Present the physical and chemical properties of the soil before planting; Also presents the same results for the treatments after harvesting.

3.     To better discuss the effects of biochar on soil microbes, I will encourage the authors to also work on soil enzymes and soil microbial diversity in their methodology as the current work load is too shallow and small to really discuss the effects of biochar

4.     Characterize the prepared biochar and also discuss the results.

Results and Discussions:

1.     The authors should use Pearson correlation to check if there is relationship among plant and soil parameters and discuss accordingly.

2.     Please significantly expand, extend and improve the quality of your discussions as the current one is shallow due to very low workload/scope, also discuss every results extensively with proper and adequate referencing.

3.     The title says “impact of interaction”, however no interactions or mechanism or interactions were discussed.

4.     Also discuss the effects of treatments on plant physiological and biochemical properties; soil physical and chemical properties like pH, EC, SOC/SOM, P, K, N, particle size distribution etc.

5.     Can cabbage of 35 days weigh about 20-30g? I doubt it.

Comments on the Quality of English Language

The English language quality is averagely ok

Round 2

Reviewer 1 Report

Comments and Suggestions for Authors

The authors have given satisfactory response and the manuscript is in good shape and can be accepted for publication.

Author Response

Original manuscript ID: applsci-2669282

Title: Impact of interaction between biochar and soil microorganisms on growth of Chinese cabbage by increasing soil fertility

Author(s): Jing Luan, Yang Fu, Wenzhu Tang, Fan Yang, Xianzhen Li and Zhimin Yu*

Dear Ms. Vukosava Milic and reviewers:

    Thank you very much for the careful reading of our manuscript entitled "Impact of interaction between biochar and soil microorganisms on growth of Chinese cabbage by increasing soil fertility", and for the opportunity to submit a revised version. We sincerely appreciate the editor and reviewers for providing valuable suggestions and comments that have greatly helped us to improve the manuscript.

    We have thoroughly considered all of the comments and have substantially revised our manuscript accordingly.

    Attached, please find our additional modifications to the journal editor / reviewers’ comment. We would be most grateful if you could consider the thoroughly rewritten manuscript for publication applied sciences.

Thank you and best regards.

Sincerely,

Zhimin Yu

School of Biological Engineering, Dalian Polytechnic University, Ganjingziqu, Dalian 116034, People’s Republic of China

Response to Reviewer 1 Comments:

Comments: The authors have given satisfactory response and the manuscript is in good shape and can be accepted for publication.

Response: We sincerely appreciate the editor and reviewers for providing valuable suggestions and comments that have greatly helped us to improve the manuscript. In Addition, we have made some modifications to the manuscript. ”a: ** p < 0.01; * p < 0.05. b: OM: Organic matter in soil; AP: Avalilable phosphorus in soil; AK: Avalilable potassium in soil; PS: Phosphatase activity of soil; US: Urease activity of soil; FW: Fresh weight of cabbage; PC: Phosphorus content of cabbage; PK: Potassium content of cabbage” was added at Table 1; “MC” was changed as “Microbial content” in Table 1; “Ochrobactrum sp.” was changed as “The relative abundance of Ochrobactrum sp.” in Table 1; “Bacillus mucilaginosus” was changed as “The relative abundance of Bacillus mucilaginosus” in Table 1.

Reviewer 2 Report

Comments and Suggestions for Authors

Dear Editor and authors, the manuscript was improved and can be accepted in the current version.

Author Response

Original manuscript ID: applsci-2669282

Title: Impact of interaction between biochar and soil microorganisms on growth of Chinese cabbage by increasing soil fertility

Author(s): Jing Luan, Yang Fu, Wenzhu Tang, Fan Yang, Xianzhen Li and Zhimin Yu*

Dear Ms. Vukosava Milic and reviewers:

    Thank you very much for the careful reading of our manuscript entitled "Impact of interaction between biochar and soil microorganisms on growth of Chinese cabbage by increasing soil fertility", and for the opportunity to submit a revised version. We sincerely appreciate the editor and reviewers for providing valuable suggestions and comments that have greatly helped us to improve the manuscript.

    We have thoroughly considered all of the comments and have substantially revised our manuscript accordingly.

    Attached, please find our additional modifications to the journal editor / reviewers’ comment. We would be most grateful if you could consider the thoroughly rewritten manuscript for publication applied sciences.

Thank you and best regards.

Sincerely,

Zhimin Yu

School of Biological Engineering, Dalian Polytechnic University, Ganjingziqu, Dalian 116034, People’s Republic of China

Response to Reviewer 2 Comments:

Comments: Dear Editor and authors, the manuscript was improved and can be accepted in the current version.

Response: We sincerely appreciate the editor and reviewers for providing valuable suggestions and comments that have greatly helped us to improve the manuscript. In Addition, we have made some modifications to the manuscript. ”a: ** p < 0.01; * p < 0.05. b: OM: Organic matter in soil; AP: Avalilable phosphorus in soil; AK: Avalilable potassium in soil; PS: Phosphatase activity of soil; US: Urease activity of soil; FW: Fresh weight of cabbage; PC: Phosphorus content of cabbage; PK: Potassium content of cabbage” was added at Table 1; “MC” was changed as “Microbial content” in Table 1; “Ochrobactrum sp.” was changed as “The relative abundance of Ochrobactrum sp.” in Table 1; “Bacillus mucilaginosus” was changed as “The relative abundance of Bacillus mucilaginosus” in Table 1.

Reviewer 3 Report

Comments and Suggestions for Authors

Accept

Author Response

Original manuscript ID: applsci-2669282

Title: Impact of interaction between biochar and soil microorganisms on growth of Chinese cabbage by increasing soil fertility

Author(s): Jing Luan, Yang Fu, Wenzhu Tang, Fan Yang, Xianzhen Li and Zhimin Yu*

Dear Ms. Vukosava Milic and reviewers:

    Thank you very much for the careful reading of our manuscript entitled "Impact of interaction between biochar and soil microorganisms on growth of Chinese cabbage by increasing soil fertility", and for the opportunity to submit a revised version. We sincerely appreciate the editor and reviewers for providing valuable suggestions and comments that have greatly helped us to improve the manuscript.

    We have thoroughly considered all of the comments and have substantially revised our manuscript.

    Attached, please find our additional modifications to the journal editor / reviewers’ comment. We would be most grateful if you could consider the thoroughly rewritten manuscript for publication applied sciences.

Thank you and best regards.

Sincerely,

Zhimin Yu

School of Biological Engineering, Dalian Polytechnic University, Ganjingziqu, Dalian 116034, People’s Republic of China

Response to Reviewer 3 Comments:

Comments: Accept

Response: We sincerely appreciate the editor and reviewers for providing valuable suggestions and comments that have greatly helped us to improve the manuscript. In Addition, we have made some modifications to the manuscript. ”a: ** p < 0.01; * p < 0.05. b: OM: Organic matter in soil; AP: Avalilable phosphorus in soil; AK: Avalilable potassium in soil; PS: Phosphatase activity of soil; US: Urease activity of soil; FW: Fresh weight of cabbage; PC: Phosphorus content of cabbage; PK: Potassium content of cabbage” was added at Table 1; “MC” was changed as “Microbial content” in Table 1; “Ochrobactrum sp.” was changed as “The relative abundance of Ochrobactrum sp.” in Table 1; “Bacillus mucilaginosus” was changed as “The relative abundance of Bacillus mucilaginosus” in Table 1.
